# Validation of the Italian version of a patient-reported outcome measure for Hereditary Spastic Paraplegia

Eleonora Diella[1]*, Maria Grazia D'Angelo[1], Cristina Stefan[2], Giulia Girardi[2], Roberta Morganti[1], Andrea Martinuzzi[3], Emilia Biffi[1]

1 Scientific Institute, IRCCS E. Medea, Bosisio Parini, Lecco, Italy, 2 Scientific Institute, IRCCS E. Medea, Pieve di Soligo, Treviso, Italy, 3 Scientific Institute, IRCCS E. Medea, Conegliano, Treviso, Italy

* eleonora.diella@lanostrafamiglia.it

**Data Availability Statement:** Data are freely available on Zenodo: https://zenodo.org/doi/10.5281/zenodo.6953521.

## Abstract

### Background and aim

Patient-reported outcome measures (PROMs) are recognized as valuable measures in the clinical setting. In 2018 we developed the Italian version of the "Hereditary Spastic Paraplegia-Self Notion and Perception Questionnaire" (HSP-SNAP), a disease-specific questionnaire that collects personal perception on motor symptoms related to HSP such as stiffness, weakness, imbalance, reduced endurance, fatigue and pain. In this study our primary aim was to assess the questionnaire validity and reliability. Our secondary aim was to characterize the symptoms "perceived" by patients with HSP and compare them with those "perceived" by age-matched healthy subjects.

### Methods

The 12-item HSP-SNAP questionnaire was submitted to 20 external judges for comprehensibility and to 15 external judges for content validity assessment. We recruited 40 subjects with HSP and asked them to fill the questionnaire twice for test-retest procedure. They also completed the Medical Outcome Survey Short Form (SF-36) and were evaluated by the Spastic Paraplegia Rating Scale and the Six-Minute Walk Test. We also recruited 44 healthy subjects who completed the HSP-SNAP once to test score variability.

### Results

The HSP-SNAP content validity index was high (0.8±0.1) and the test-retest analysis showed high reliability (ICC = 0.94). The mean HSP-SNAP score (score range 0–48) of the HSP group was 22.2±7.8, which was significantly lower than healthy subjects (43.1±6.3). The most commonly perceived symptom was stiffness, followed by weakness and imbalance.

### Conclusion

Although HSP-SNAP does not investigate non-motor symptoms and we validated only its Italian version, it showed good validity and reliability and it could be used in combination with

**Funding:** This work was supported by Fondazione Cariplo and Regione Lombardia (EMPATIA@Lecco project) and by the Italian Ministry of Health (RC2022/2024 to E. Biffi). There was no additional external funding received for this study.

**Competing interests:** The authors have declared that no competing interests exist.

**Abbreviations:** PROMS, Patient-Reported Outcome Measure; HSP, Hereditary Spastic Paraplegia; HSP-SNAP, Hereditary Spastic Paraplegia Self- Notion and Perception; ICC, Intra-Class Correlation; SPRS, Spastic Paraplegia Rating Scale; MMT, Manual Muscle Testing; 6MWT, Six-Minute Walk Test; MAS, Modified Ashworth Scale; TUG, Timed Up and Go test; 10MWT, Ten Meters Walk Test; SCA, Spino-Cerebellar Ataxia; SF-36, Short-Form-36; CVR, Content Validity Ratio; CVI, Content Validity Index; SEM, Standard Error of Measurement; MDC, Minimum Detectable Change; SD, Standard Deviation.

other objective outcome measures for clinical purposes or as endpoints for future clinical rehabilitation studies.

## Trial registration

Trial Registration: ClinicalTrial.gov, NCT04256681. Registered 3 February 2020.

## Background and purpose

Hereditary spastic paraplegia (HSP) is a rare genetically and clinically heterogeneous group of neurodegenerative disorders characterized by gait disturbance mainly due to spasticity and weakness of the lower limbs [1]. HSP (prevalence 1.8–9.8/100.000) is characterized by progressive pyramidal symptoms due to primary retrograde degeneration of the corticospinal axonal fibers [2], and disease onset can occur at any age (congenital, pediatric and adult onset). Traditionally, HPS patients are classified into "pure" and "complicated" forms, depending on additional features besides pyramidal signs, according to Appleton and colleagues [3]. "Pure" forms are characterized by lower limb spasticity and weakness, hyperreflexia, extensor plantar responses, partial distal loss of vibration sense and urinary urgency [4]. "Complicated" forms could also include cerebellar ataxia, peripheral neuropathy, retinopathy, epilepsy, and cognitive impairment [5]. Spastic gait is the clinical core feature in patients with HSP, with spasticity, muscle co-activation and progressive weakness contributing to worsening motor performance and increased energy consumption [6]. In addition, postural instability causes reduced endurance, increased fatigue and increased risk of falls [7,8]. Slow functional decline can lead to limitations in autonomy and quality of life [9].

So far, no effective therapy can prevent, delay or reverse nerve degeneration in HSP and treatments are mainly directed at symptoms control. Antispastic medications and physiotherapy may help reduce functional gait impairment; nevertheless, evidence-based guidelines or recommendations are not available yet [10,11]. According to a recent review on the management of HSP [12], there are few and low-quality studies on physical treatment. At the same time, there is a lack of RCTs with larger sample size and comparable as well as validated outcome measures to evaluate the efficacy of innovative or traditional treatment. The lack of multicenter studies of its natural history is a major limitation to more targeted treatment indications for HSP. There is thus an urgent need for specific clinical assessment measures with strong psychometric properties, namely validity, reliability and sensitivity [13].

The Spastic Paraplegia Rating Scale (SPRS), the Manual Muscle Testing (MMT) and the Six-Minute Walk Test (6MWT) are recommended for the characterization of the HSP population [14–16]. Other commonly used measures in clinical trials recruiting patients with HSP include the Modified Ashworth Scale (MAS), the Timed Up and Go Test (TUG) and the Ten-Meters Walk Test (10MWT). Given this broad range of objective measures, patient-reported outcome measures (PROMs) provide patients' viewpoint about symptoms' severity, functional and psychological problems, treatment satisfaction and quality of life, thus complementing objective outcome measures. PROMs have been used as outcomes in clinical trials for many rare diseases and could be useful for monitoring changes in disease natural history and disease progression [17–19]. In his 8-year follow-up of 526 patients affected by spinocerebellar ataxia (SCA) with different PROMs, Jacoby observed that patient-reported outcome measures could detect changes during disease progression [20]. Generic PROMs are comparable across diseases but may miss important population-specific data as well as lack sensitivity and

responsivity to disease specificities vs PROMs tailored to a specific rare condition [21]. As Amprosi recently undelined, generic available PROMs are not suitable for detecting changes in HSP patients, and thus, he proposed to create a tailored HSP-PROM that could be a meaningful integration to standard functional clinical outcomes [22].

In 2018 we developed the "Hereditary Spastic Paraplegia-Self Notion and Perception Questionnaire" (hereinafter, HSP-SNAP), a disease-specific HSP-PROM, with the aim of collecting the patient's personal evaluation of specific motor symptoms that can impact on walking ability.

The first aim of this study was to assess the comprehensibility, content validity, reliability and correlation of HSP-SNAP with other selected outcome measures.

Our second aim was to characterize the HSP sample recruited through the analysis of each single dimension explored by the questionnaire. We hypothesized that stiffness was the symptom most commonly perceived by patients, probably followed by weakness and imbalance and, finally, by reduced endurance, pain and fatigue. Furthermore, we compared symptoms "perceived" by HSP patients and those "perceived" by healthy subjects to collect data on the variability of scoring in a healthy population.

## Materials and methods

### HSP-SNAP

The HSP-SNAP was developed in 2018 by two physical therapists experienced in motor disorders and neurological rehabilitation working for 20 years with patients affected by HSP, and a bioengineer at Scientific Institute E. Medea in Bosisio Parini, Italy. The HSP-SNAP is a patient-reported measure focusing on 6 key features for HSP: stiffness, weakness, imbalance, reduced endurance, fatigue and pain. All these motor symptoms are regularly mentioned by patients, assessed by clinicians and reported in the literature [6,7,14,23,24]. The aim of the HSP-SNAP is to assess the subjective impact of these symptoms on gait and walking performance.

The tool consists of 12 items, 2 items for each dimension. Specifically, stiffness is assessed by items 1 and 6, weakness by items 3 and 8, imbalance by items 9 and 12, reduced endurance by items 4 and 7, fatigue by items 2 and 11 and pain by items 5 and 10. The two items of each dimension are balanced in terms of positive/negative attitude to the question, to avoid "automatic compiling". For example, stiffness is analyzed by item 1 "My leg stiffness affected my walking" (negative attitude towards stiffness) and item 6: "I felt my walking smooth/unrestrained" (positive attitude towards stiffness). The HSP-SNAP uses a 5-point Likert scale where 1 = strongly disagree, 2 = disagree, 3 = neutral, 4 = agree, 5 = strongly agree.

To correctly weigh items with positive and negative attitude, scores are computed as follows: the score contribution of items with a positive attitude (i.e. item 2, 4, 6, 8, 10, 12) is the "score rated by the patient minus 1" (pt score–1); the contribution of items with a negative attitude (i.e. items 1, 3, 5, 7, 9, 11) is "5 minus the score rated by the patient" (5–pt score)". The total score is the sum of all item scores, with a maximum score of 48 (score range is 0–48) while the dimension score is the mean value between the two items related to that dimension. The higher the score, the greater the individual well-being and the milder the symptoms.

Table 1 shows, as a reference, the HSP-SNAP translated in English by a native speaker to be understandable by international readers. A proper translation process including translation and back-translation by an independent group was performed for the English version. The Italian version, validated in this manuscript, is provided as (S1 Table).

**Table 1. English translation of the original Italian version of HSP-SNAP questionnaire.**

| Self-Notion and Perception Questionnaire in Hereditary Spastic Paraplegia (HSP-SNAP) | | | | |
|---|---|---|---|---|
| **Choose an answer for each statement.** | | | | |
| Strongly disagree 1 | Disagree 2 | Neutral 3 | Agree 4 | Strongly agree 5 |
| In the last week: | | | | |
| **1. My leg stiffness affected my walking.** | | | | |
| 1 | 2 | 3 | 4 | 5 |
| **2. I carried out my daily motor activities without physical fatigue.** | | | | |
| 1 | 2 | 3 | 4 | 5 |
| **3. My leg weakness affected my walking.** | | | | |
| 1 | 2 | 3 | 4 | 5 |
| **4. I felt I had good resistance while walking.** | | | | |
| 1 | 2 | 3 | 4 | 5 |
| **5. Physical pain hindered me in my daily motor activities.** | | | | |
| 1 | 2 | 3 | 4 | 5 |
| **6. I felt my walking smooth/unrestrained.** | | | | |
| 1 | 2 | 3 | 4 | 5 |
| **7. My poor resistance during walking hindered me.** | | | | |
| 1 | 2 | 3 | 4 | 5 |
| **8. I felt my legs strong while walking.** | | | | |
| 1 | 2 | 3 | 4 | 5 |
| **9. I felt I had poor balance while walking.** | | | | |
| 1 | 2 | 3 | 4 | 5 |
| **10. I carried out my daily motor activities without feeling physical pain.** | | | | |
| 1 | 2 | 3 | 4 | 5 |
| **11. The physical effort limited my daily activities.** | | | | |
| 1 | 2 | 3 | 4 | 5 |
| **12. I felt I had good balance while walking.** | | | | |
| 1 | 2 | 3 | 4 | 5 |

## Participants

The study was approved by the local ethics committee (GIP 595; date of approval 9/20/2018). We enrolled the first patient on the 30th October 2018 and we concluded the study on the 31th May 2021. The trial was registered on Clinicaltrial.gov (NCT04256681) after the first patient's recruitment but no changes were done to the protocol during the study. Informed written consent was signed by all study participants or by their parents.

**External judges.** External judges were recruited to assess HSP-SNAP comprehensibility and content validity. The evaluation panel for the comprehensibility assessment included a group of clinical experts and two small groups of patients with HSP and healthy subjects. For the content validity assessment, we selected a panel of clinical experts and patients with HSP, according to Lawshe [25,26].

**Patients with HSP.** In this multicenter study, patients with HSP were recruited at Scientific Institute E. MEDEA in Bosisio Parini (leading center) and in Pieve di Soligo (Italy) according to the following diagnostic clinical criteria: 1) "pure" or "complicated" genetically determined HSP or 2) not genetically determined paraplegia with the exclusion of all other

causes of spasticity through brain and spinal cord imaging studies, metabolic analysis, clinical history data (i.e. pregnancy and or/ delivery complication, Apgar evaluation at birth).

Inclusion criteria were: age above 9 years (which is considered the minimum age at which subjective symptoms can be critically evaluated), ability to walk at least 10 meters (a walking device was allowed). Exclusion criteria included: wheelchair-bound patients, patients with severe orthopedic conditions, cardiovascular dysfunction, psychopathological symptoms, moderate/severe mental retardation or cognitive decline.

To estimate the sample size, the agreement between two measures of HSP-SNAP on two consecutive days (see Study Protocol) was selected as primary outcome measure. Given an acceptable reliability level of 0.6, an expected reliability of 0.8, a power of 0.80 with alpha = 0.05, it was estimated that the number needed to verify the agreement was 39 patients [27,28]. G*Power 3.1.9.4 was used to calculate the sample size.

**Healthy subjects.**   Healthy subjects were recruited according to the following inclusion criteria: age above 9 years and being in good health.

## Study protocol

We designed our protocol following the COSMIN guidelines for studies on measurement properties of patient reported outcome measures. The Cosmin Checklist is provided as (S1 Checklist).

The HSP-SNAP was evaluated by two panels of judges. For the comprehensibility assessment, we asked to 20 judges to verify if the items describe the competency explored in a complete and effective way, both from a structural and linguistic point of view. Each item could be judged as totally comprehensible, partially comprehensible or not comprehensible.

For the content validity assessment, we conducted short semi-structured interviews with 15 judges for their personal point of view on the HSP-SNAP symptoms' selection. Each item could be valued as essential/necessary, useful but not essential, or not essential/necessary [25,29].

Afterwards, during a single-day evaluation at the hospital, patients were examined by experienced physical therapists by the Spastic Paraplegia Rating Scale (SPRS) and, if possible, the Six Minute Walking Test (6MWT). The SPRS evaluates the multisystem involvement of HSP, covering in 13 items the wide spectrum of different aspects of the disease to evaluate its severity (15). The 6MWT is used to evaluate walking endurance; it measures the walking distance covered in six minutes along a 25 m standardized path (16). Additionally, patients completed the SF-36 and the HSP-SNAP. The SF-36 is the most extensively validated and used outcome survey for quality of life. It provides a global view of health and is one of the most commonly used general PROM [30,31]. It is made by 36 items that can be aggregated in 8 sub-scales and in 2 summary measures, i.e. physical component and mental component of health, according to Apolone et al [32].

The HSP-SNAP was self-administered via paper and pencil. No time constraint was given to the patients. Participants received a clear explanation by the evaluator and could ask for clarifications. The evaluator used standardized instructions to avoid bias introduction. Furthermore, we recorded potential difficulties in understanding and completing each single item.

The HSP-SNAP was completed by patients with HSP a second time on the following day to assess its reliability. We ensure no environmental changes occurred between the two assessments. Participants were not aware that this was a test-retest evaluation and we explicitly stated that the second test was designed to evaluate their current status.

Healthy subjects filled the HSP-SNAP form once with the purpose to define normative data.

## Statistical analysis

**HSP-SNAP comprehensibility and content validity, reliability and correlation.** To assess item comprehensibility, for each item we determined to what percentage of judges declared the item totally comprehensible.

To assess the *content validity ratio (CVR)*, the data collected by judges were analyzed item-by-item according to Eq 1 [25]:

$$\text{CVR} = \frac{n_{e-N/_2}}{N/_2} \tag{1}$$

where $n_e$ is the number of judges judging items "essential" and N is the total number of judges.

The CVR was negative when less than half of the judges answered "essential"; the CVR = 0 when 50% of judges answered "essential" and the CVR = 1 when all judges answered "essential".

We then computed the *content validity index* (CVI) for the whole test, simply calculating the mean CVR values of items.

The normality of the distribution was assessed for all the variables by means of Shapiro-Wilk test and the following analyses were defined accordingly.

The HSP-SNAP reliability was assessed for possible systematic errors by means of a paired T-test and calculating the intraclass correlation coefficient (ICC) for absolute agreement between the two measures collected on two consecutive days in the 40 subjects with HSP. Furthermore, the Standard Error of Measurement (SEM) and the Minimum Detectable Change (MDC) were computed, according to Polit [33]. The Pearson correlation was computed between the two measures of HSP-SNAP.

To verify that the presence of underaged in the recruited group does not affect the reliability of HSP-SNAP, we performed the analysis including only adults.

Furthermore, potential relations between the HSP-SNAP and SPRS, 6MWT and SF-36 (physical component and mental component) were explored by Pearson correlation.

**Characterization of the HSP sample.** The HSP group was compared with the healthy group in terms of age (by means of a Mann-Whitney U test), and gender (Chi-squared test). The mean HSP-SNAP scores were computed in the two groups and compared by means of a Mann-Whitney U test. Furthermore, the correlation between the HSP-SNAP scores and age was assessed by means of Spearman rank correlation in the two groups. The possible influence of gender was assessed with a Chi-squared test. In the HSP group, the correlation of HSP-SNAP with age at disease onset was further evaluated. To look for potential differences between pure and complicated HSP, a Mann-Whitney U test was applied to compare HSP-SNAP scores in patients with pure and complicated HSP.

Finally, we analyzed each of the six dimensions (i.e. stiffness, weakness, imbalance, reduced endurance, fatigue and pain) explored by the questionnaire. We computed the mean between the two items related to the same dimension and compared them by means of a repeated measure ANOVA and Bonferroni-corrected T test as post-hoc analysis. To conclude, we computed the Pearson correlation between the total HSP-SNAP score and the six dimensions.

# Results

## Participant description

20 judges were recruited to assess the comprehensibility of HSP-SNAP. The judges were 10 clinicians (physical therapists; work experience on average 15 years), 5 patients with HSP and 5 healthy subjects (Table 2, top).

**Table 2. Description of the judges recruited in the study.** We both reported Comprehensibility (top) and Content Validity (bottom) judges informations.

| COMPREHENSIBILITY | | | |
|---|---|---|---|
| | HSP patients | Healthy subjects | Expert clinicians |
| Number | 5 | 5 | 10 |
| Gender, M/F | 4/1 | 1/4 | 4/6 |
| Age, y (mean ± SD) | 30.8±14.2 | 35.2±20.2 | 37.9±11.7 |
| Age at onset, y (mean ± SD) | 11.4±14.6 | - | - |
| Molecular diagnosis | 3/2 | - | - |
| Pure/complicated HSP | 3/2 | - | - |
| CONTENT VALIDITY | | | |
| | HSP patients | Healthy subjects | Expert clinicians |
| Number | 5 | - | 10 |
| Gender, M/F | 5/0 | - | 0/10 |
| Age, y (mean ± SD) | 34.2±25.5 | - | 37.7±10 |
| Age at onset, y (mean ± SD) | 20.8±22.4 | - | - |
| Molecular diagnosis | 3/2 | - | - |
| Pure/complicated HSP | 2/3 | - | - |

15 judges were recruited to assess content validity of HSP-SNAP. The judges were 10 expert clinicians (specialized neurologists, specialized physiatrists and physical therapists; work experience on average 14 years), and 5 patients affected by HSP (Table 2, bottom).

Table 2 reports the characteristics of the judges for both comprehensibility and content validity assessment. According to Lawshe, the required minimum value of CVR was 0.49, considering 15 judges.

Table 3 reports the characteristics of the participants. 40 patients with HSP were recruited according to the sample size estimation. The molecular diagnosis was available in 24 of 40 patients (10 were carrying mutation in SPG4 (9 pure and 1 complicated); 4 in SPG7 (complicated); 2 in SPG3A (pure); 2 in SPG11 (complicated); 1 in SPG5 (pure); 1 in SPG8 (complicated); 1 in SPG30 (pure); 1 in SPG31 (pure); 1 in SPG35 (complicated); 1 in SPG72 (pure). 27 patients were affected by a "pure" HSP, whereas 13 patients were affected by a "complicated" form (with an addition of other neurological and non-neurological symptoms, such as cerebellar ataxia, peripheral neuropathy, retinopathy and mild cognitive impairment). 4 out of 40

**Table 3. Description of the population included in the study.**

| | HSP patients | Healthy subjects | P value* |
|---|---|---|---|
| Number | 40 | 44 | |
| Gender, M/F | 27/13 | 19/25 | .025 |
| Age, y (mean ± SD) | 45.4 ± 17.8 | 42.3 ± 19.6 | .337 |
| Age at onset (y) | 35.8 ± 21.3 | - | |
| Molecular diagnosis | 24/40 | - | |
| Pure/complicated HSP | 27/13 | - | |
| SPRS (mean ± SD) | 16.8 ± 7.7 | | |
| 6MWT, meters (mean± SD) | 299.9 ± 106.8 | | |
| SF36 (mean± SD) | 98.7 ± 5.4 | | |

SD: Standard deviation

* These p-values refer to the comparison between patients with HSP and healthy subjects.

patients were showing a mild mental retardation but had the ability to critically answer to the questions. 4 out of 40 patients were regularly taking antispastic medication. 3 out of 40 were taking antidepressant drugs.

44 healthy subjects were enrolled in the study to have a sample size comparable to that of patients with HSP. The two groups (healthy subjects vs subjects with HSP) were comparable in terms of age (p = .468) but not in terms of gender (p = .025), being the group of patients with a prevalence of males.

## Primary outcome

At first, we evaluated item comprehensibility which was very high, with values higher than 0.9.

Then, we measured item content validity by Eq 1 according to Lawshe. The CVR was 0.87 for four items; 0.73 for six items and 0.6 for two items. These values show good validity for each item considered complete, clear and effective and allowed us to use the original version of the HSP-SNAP in the experimental phase. The CVI of the HSP-SNAP was 0.8±0.1.

Table 4 shows the CVR value for each item and the ratio between the number of judges that declared the items were clear over the total number of judges (Com).

All the 40 patients with HSP completed the HSP-SNAP questionnaire twice. No missing items were detected. The time required for compilation was approximately 3–5 minutes, with a maximum of 10 minutes.

We then performed the reliability analysis. No systematic errors between the two measures emerged on the T-test (p = .476; t -.720). Furthermore, with an ICC = 0.94, the HSP-SNAP showed an optimal test-retest reliability. The SEM was 1.7, which means that approximately two points in the HSP-SNAP were due to the scale unreliability. The MDC was 4.8. The two measures of HSP-SNAP showed a good correlation (r = .883, p < .001). We also looked for possible correlations with other outcome measures. The mean values of SPRS, 6MWT (available for 35 patients) and SF-36 (available for 34 patients) are reported in Table 3. Pearson's correlation analysis showed no correlation between the HSP-SNAP and SPRS (r = - .174, p = .284) and 6MWT (r = .149, p = .393). On the other hand, a moderate correlation between HSP-SNAP and the physical component (r = .549, p = .001, Fig 1) and the mental component (r = .370, p = .031) of SF-36 emerged.

The results about the reliability analysis considering only the adult group (N = 35) are reported as (S1 File).

**Table 4. Comprehensibility and content validity ratio and of each of the 12 items.**

| Item | Com | CVR |
|------|-----|-----|
| 1 | 1 | 0.87 |
| 2 | 0.95 | 0.73 |
| 3 | 1 | 0.73 |
| 4 | 0.95 | 0.73 |
| 5 | 1 | 0.60 |
| 6 | 0.9 | 0.87 |
| 7 | 1 | 0.87 |
| 8 | 0.95 | 0.60 |
| 9 | 1 | 0.87 |
| 10 | 1 | 0.73 |
| 11 | 0.95 | 0.73 |
| 12 | 1 | 0.73 |

Com: Comprehensibility, CVR: Content validity ratio.

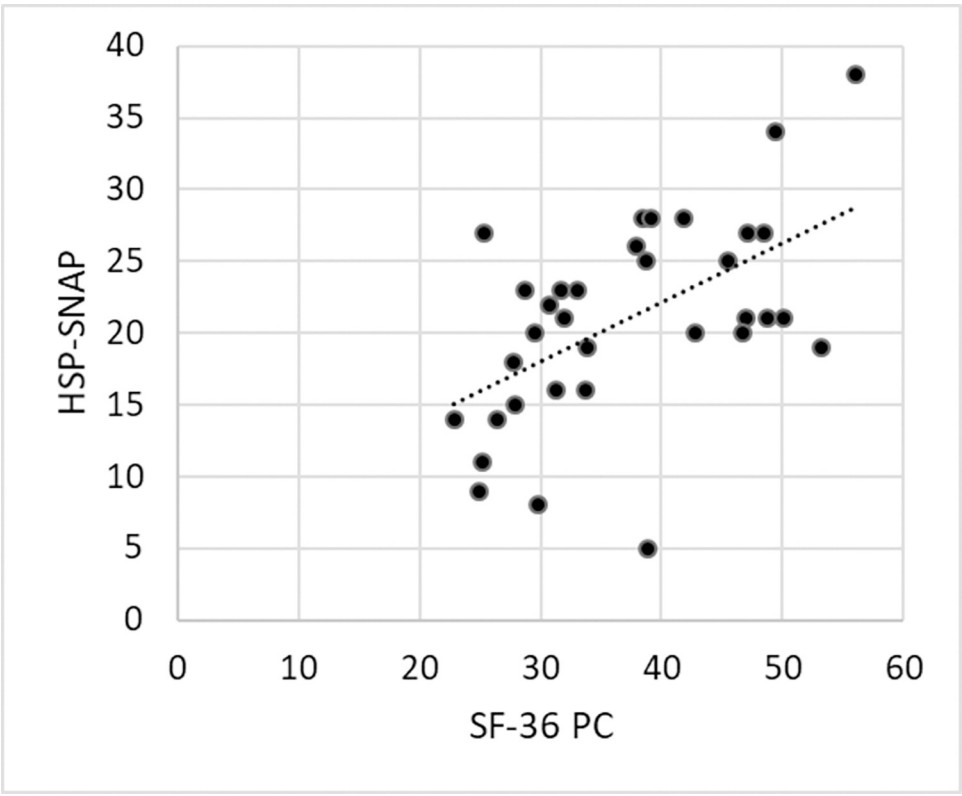

**Fig 1. Correlation between HSP-SNAP and SF-36 physical component (PC) in the group of patients with HSP.**
The dashed line represents the correlation line.

### Secondary outcome

All healthy subjects filled the HSP-SNAP questionnaire once. No missing items were observed. The mean and standard deviation of HSP-SNAP scores in the healthy population and in the HSP group were 43.1±6.3 and 22.2±7.8, respectively, showing a statistically significant difference (U = 36.5, p < .001). In both groups, the HSP-SNAP was significantly correlated with age (rho = -.548, p < .001 in healthy subjects, rho = -.362, p = .022 in HSP subjects) but not influenced by gender (p = .602 in healthy subjects, p = .565 in subjects with HSP). In the HSP group, the HSP-SNAP was not correlated with age at disease onset (rho = -.274, p = .087). The HSP-SNAP scores in patients with pure and complicated HSP were 20.8 ± 7.4 and 20.9 ± 7.3, respectively, and they did not differ significantly (p = 0.080).

As far as HSP-SNAP items are concerned, the repeated measure ANOVA with Bonferroni correction for multiple comparison highlighted several differences between dimensions (Fig 2; to improve readability, items are shown from the less disabling symptom (top, longest bar) to the most disabling one (bottom, shortest bar)). As expected, the most commonly perceived symptom was stiffness (Fig 2, blue bar) showing the lowest values with a statistically significant difference vs fatigue (p = 0.002, Fig 2, orange bar, horizontal line pattern), reduced endurance (p = 0.006, Fig 2, green dotted bar) and pain (p<0.001, Fig 2, violet bar, vertical line pattern). Weakness (Fig 2, grey checkered bar) was perceived as debilitating, with significantly lower values than fatigue (p = 0.031) and pain (p<0.001). The third most commonly perceived symptom was imbalance (Fig 2, pink bar, diagonal line pattern), with significantly lower values than pain (p = 0.004). Finally, low resistance, too, was perceived as more disabling than pain

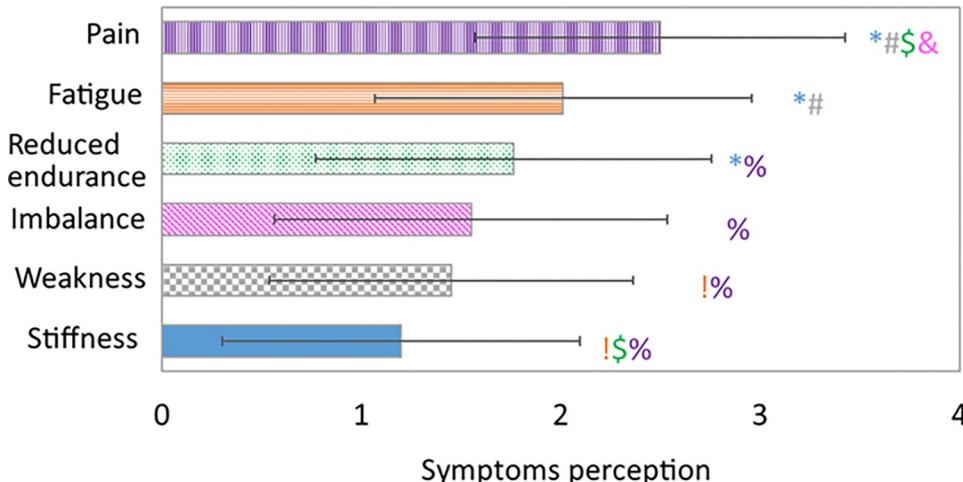

**Fig 2. HSP-SNAP dimensions as scored by the 40 patients with HSP.** Mean and standard deviation values of HSP-SNAP dimensions are reported. Symbols show statistically significant differences in repeated measure ANOVA with Bonferroni correction for multiple comparisons. Blu *: Difference among stiffness and the other dimensions. Grey #: Difference among weakness and the other dimensions. Pink &: Difference among imbalance and the other dimensions. Green $: Difference among reduced endurance and the other dimensions. Orange!: Difference among fatigue and the other dimensions. Violet %: Difference among pain and the other dimension.

(p = 0.004), which was the least perceived symptom. All these symptoms well correlated with the total HSP-SNAP score (rho from .368 to .755, all p< 0.02).

## Discussion

In recent years, the need for personalized treatment approaches stressed the importance of knowing the patients' viewpoint as a core aspect for targeted treatment. This is usually done by patient-reported outcome measures and the literature is full of PROMs focusing on general health and well-being which however are not sensitive to the issues that patients with HSP face. In the field of spinocerebellar diseases, Schmahmann [34] developed and validated an ataxia-specific PROM in 2021 to assess patients' perception of balance but this was not comprehensive of all motor symptoms reported by patients with HSP. We thus decided in 2018 to develop the "Hereditary Spastic Paraplegia-Self Notion and Perception Questionnaire" (HSP-SNAP) designing the questionnaire with clinicians with a long-standing knowledge of patients' perspective. The end users were involved in its comprehensibility and content validity assessment.

Results showed optimal comprehensibility and globally good scores of content validity ratio (CVR) for each single item and a high CVI score, confirming a good total content validity. The judges judged the questionnaire as complete and did not suggest including any other items.

The HSP-SNAP test-retest reliability and the high test-retest reliability value (ICC) confirmed that the HSP-SNAP is a robust measure of subjective symptom perception in the HSP population. We also tested the HSP-SNAP correlation with SPRS, 6MWT and SF-36 and, as expected, we did not find any correlation either with SPRS or with 6MWT. Instead, a moderate correlation with both physical and mental component of SF-36 emerged. The lack of correlations with both SPRS and 6MWT is not surprising, given the way the questionnaire was designed and the different dimensions tapped. The SPRS describes the multisystem

involvement of HSP, covering a wider spectrum of different aspects of the disease to evaluate its severity; even if some items are slightly overlapping with the HSP-SNAP dimensions, the approach (objective vs subjective) is different. In addition, the 6MWT investigates patient endurance in a walking timed test whose result is obviously affected by symptoms. Being a PROM, the HSP-SNAP is more comprehensive than a specific functional test.

The lack of correlation is not a limitation. It provides further evidence that a patient's point of view may be different from what is usually seen in a clinical setting and must be considered along clinical objective measures. Our questionnaire highlights how objective and subjective evaluations can differ in chronic neurologic disorders and underlines the relevance of patient perception. A patient's "body" can feel earlier than an objective measure–and at the same time to a greater extent–any changes that an objective evaluation needs time to show, as it is more sensitive to changes. In 2021, Maas [35] explored the correlation between patient-reported and clinician-based outcomes in a study on 20 early-to-middle stage SCA 3 patients, underlining the divergence between these measures evaluating distinct aspects of disease. This also emphasizes their complementariness in therapeutic trials. Ruther [36] susstated that PROMs should be considered in addition to other approved methods, and not as their substitute, especially in studies with small sample sizes as it is often the case with rare diseases.

The moderate correlation between HSP-SNAP and SF-36, which collects subjective evaluations of both motor functional impairments and psychological well-being, supports the fact that the HSP-SNAP measures the subjective impact of HSP-related symptoms. This global dimension consists of elements reflecting patients' perception of the disease and may be influenced by personal perspectives, feelings and moods as sensitivity, resilience, self-esteem, motivation and emotional well-being [19]. According to the International Classification of Functioning, Disability and Health (ICF) framework, the HSP-SNAP evaluates the influence of personal background on patient's health and performance [37].

Concerning the second purpose of this manuscript, we found no correlations between the HSP-SNAP score and patients' gender or age at onset as expected by clinical experience. On the other hand, the moderate correlation with age characterizes both patients and healthy subjects and may be justified by worsening subjective perception due to aging. The comparison of HSP-SNAP scores between "pure" and "complicated" forms did not show statistical differences.

Concerning the evaluation of the HSP-SNAP subscales, we hypothesized that the symptoms mainly perceived as disabling would be stiffness, weakness and imbalance. Our results support our assumption, showing that the primary aspects of this disease perceived by patients is stiffness, resulting in lack of gait smoothness, followed by lack of strength and impaired balance. On the other hand, reduced endurance, fatigue and pain seem to be perceived as less disabling in walking. However, we are aware of the ongoing debate in the field whether HSP's gait is actually characterized by reduced endurance or just by reduced walking speed. This is a crucial open point that need further in-depth analysis. Besides, we can't ignore that fatigue and pain, as reported by Rattay et al [38], are frequent symptoms that need to be screened and treated because they affect health-related quality of life. Finally, we compared "perceived" symptoms in patients with HSP group and in healthy subjects. Data showed a statistically significant difference between scores in these two populations, with healthy subjects scoring twice as much as HSP subjects. This proves the sensitivity of the HSP-SNAP in measuring changes due to pathology. Only healthy subjects reached the maximum score of 48 points, whereas in the HSP population, HSP-SNAP seems to be relevant in all disease stages with an almost normal distribution without score of 0 and 48 points preventing a floor and ceiling effect.

This work has some limitations. First of all, patients' heterogeneity: a broad spectrum of different genotypes and phenotypes of HSP patients was included in our analysis. One third of

our HSP population had a complicated HSP with signs and symptoms different from those tapped in the HSP-SNAP, such as visual impairment or peripheral neuropathy that could further affect patients' walking ability. Nevertheless, being HSP a relatively rare disease, we believe that 40 patients make a good sample size.

Additionally, we did not collect specific information on mood or anxiety disorders potentially influencing the assessment and we assumed that the presence of depressive symptoms or symptoms of anxiety could be correlated with systematically lower scores of the instrument.

Despite these limitations, this study has several strengths. The HSP-SNAP is the first assessment measure, validated in its Italian version, that is comprehensive of all the most significant motor signs and symptoms that could affect walking ability in the HSP population. It is a short questionnaire that can be completed in less than 10 minutes and does not need any special setting or equipment. Moreover, it focuses on the patient as stakeholder in the evaluation process.

## Conclusions

The HSP-SNAP questionnaire can be regularly used in combination with other objective outcome measures either for clinical purposes or as endpoints for future clinical rehabilitation studies, to ensure that individuals with HSP receive proper help for their needs. The HSP-SNAP may also support communication between patients and clinicians as well as shared decision-making in the rehabilitation process. Future studies should test the English version of the HSP-SNAP in an English-speaking HSP population and may help validate the HSP-SNAP in clinical studies with independent and larger patient cohorts in view of a wider clinical use.

## Supporting information

**S1 Checklist. COSMIN checklist for studies on measurement properties of patient reported outcome measures.**
(DOC)

**S1 Table. Italian version of the HSP-SNAP.**
(DOCX)

**S1 File.**
(DOCX)

**S2 File.**
(DOCX)

**S1 File. Results related to the analysis of HSP-SNAP reliability considering the adult group.**
(DOCX)

## Acknowledgments

The authors wish to thank all participants.

## Author Contributions

**Conceptualization:** Eleonora Diella, Roberta Morganti, Emilia Biffi.

**Data curation:** Eleonora Diella.

**Formal analysis:** Emilia Biffi.

**Investigation:** Eleonora Diella, Cristina Stefan, Giulia Girardi.

**Methodology:** Eleonora Diella.

**Software:** Emilia Biffi.

**Supervision:** Eleonora Diella, Maria Grazia D'Angelo, Andrea Martinuzzi, Emilia Biffi.

**Validation:** Emilia Biffi.

**Writing – original draft:** Eleonora Diella.

**Writing – review & editing:** Eleonora Diella, Maria Grazia D'Angelo, Emilia Biffi.

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
