## [Decision Letter · Decision Letter 0]

28 Nov 2023

PONE-D-23-28019Development and Validation of a Patient-Reported Outcome Measure for Hereditary Spastic ParaplegiaPLOS ONE

Dear Dr. diella,

Thank you for submitting your manuscript to PLOS ONE. After careful consideration, we feel that it has merit but does not fully meet PLOS ONE’s publication criteria as it currently stands. Therefore, we invite you to submit a revised version of the manuscript that addresses the points raised during the review process.

We look forward to receiving your revised manuscript.

Kind regards,

Abeer El Wakil, PhD

Academic Editor

PLOS ONE

Journal Requirements:

“This work was partially supported by Fondazione Cariplo and Regione Lombardia (EMPATIA@Lecco project) and by the Italian Ministry of Health (RC2022 to E. Biffi).”

“The authors have no relevant financial or non-financial interests to disclose.”

4. Please include captions for your Supporting Information files at the end of your manuscript, and update any in-text citations to match accordingly. Please see our Supporting Information guidelines for more information: http://journals.plos.org/plosone/s/supporting-information. .

Reviewers' comments:

Reviewer's Responses to Questions

**Comments to the Author**

1. Is the manuscript technically sound, and do the data support the conclusions?

Reviewer #1: Yes

Reviewer #2: Partly

2. Has the statistical analysis been performed appropriately and rigorously? 

Reviewer #1: Yes

Reviewer #2: Yes

3. Have the authors made all data underlying the findings in their manuscript fully available?

Reviewer #1: No

Reviewer #2: No

4. Is the manuscript presented in an intelligible fashion and written in standard English?

Reviewer #1: Yes

Reviewer #2: Yes

5. Review Comments to the Author

Reviewer #1: In their research article „Development and Validation of a Patient-Reported Outcome Measure for Hereditary Spastic Paraplegia“ Diella et al. present the evaluation of a novel questionnaire to capture HSP specific quality of life, including rating by „external judges“, test-retest-reliability, and a first application in a mixed cohort of 40 HSP patients. As outlined in the manuscript, strengths of this study include a formal evaluation process as well as recruitment of a considerable patient cohort. The study is well conducted, the data are sound and of high interest for future trials in HSP. However, several issues need to be resolved.

1. First, „development“ of the questionnaire seems overstated since it had been developed before and was evaluated (and not modified) in this manuscript. My suggestion is to omit this term from the title, abstract and main text.

2. In addition, the term HSP-SNAP does not reflect that it probably does not reflect motor-independent symptoms like sensory symptoms, bladder dysfunction etc. which should be stated in the abstract.

3. Abstract: The „Medical Outcome Survey Short Form“ should be specified by „(SF-36)“. State absolute range of the proposed HSP-SNAP in the abstract. Note the language of the questionnaire.

4. Data availability statement: data are not freely available, but have to be requested via the given repository. This should be mentioned in the statement.

5. Study registration in ClinicalTrials is credited, but a closer look shows that the study design was posted 1.5 years after FPI; this limitation must be mentioned in the methods as well as any changes to the protocol during the study.

6. An average of 10 minutes to complete a 12-item-questionnaire seems odd. This might have been caused by „exhaustive explanations by the evaluator“ which however may introduce substantial bias and must be discussed.

7. The study included pediatric and adult patients, with an inclusion starting at age 8. Given the clinical differences of pediatric onset HSP and the differences in PROM responses provided by pediatric cohorts, the authors should precisely state the ratio of minors and provide results of their statistical analyses when including adults only as a supplement.

8. Ref. 18 seems out of context.

9. Ll. 131-146 do not belong to the „background & purpose“ section – they should be replaced by a much more concise summary of overall hypothesis and findings.

10. „Unbalance“ should be rephrased throughout the manuscript.

11. L. 278: please provide the reference.

12. It is surprising that expert clinicians were different for assessing comprehensibility and content validity, and a reason should be provided. For the „expert clinicans“ group, year of experience with HSP patients should be reported.

13. Table 2 lacks SD data (as indicated in the first row), and SD data should also be reported for „age at onset“.

14. The „COSMIN Reporting guideline for studies on measurement properties of patient reported outcome measures“ provided as a supplement is the unmodified raw document by PMID33818733. For each of the items on this list, the authors must provide a statement how this item was met in the present study.

15. The Italian questionnaire version that was acutally applied should be provided as a supplement.

16. The figure legends should be added to the submission file, including abbreviations.

17. The exact version of the SF-36 questionnaire should be specified. Moreover, the method of calculating SF-36 values for correlational analyses is unclear (physical or mental component scores? type of calculation).

18. l. 329: Define whether SD or SEM is reported here.

19. Fig. 2: The calculation of the x-axis value should be made clear to the reader in the figure legend. The color coding presentation of significance is unusual and not distinguishable on printing b/w. The smileys suggest that they were present on the actual questionnaires.

20. Typos must be corrected here: l. 84, l. 110, l. 114, l. 118 (twice), l. 119 (twice), l. 139, l. 286, l. 365, l. 408

Reviewer #2: With interest I read the submitted manuscript “Development and Validation of a Patient-Reported Outcome Measure for Hereditary Spastic Paraplegia” by Diella and colleagues.

Major:

I see severe methodological issues with this “patient reported” outcome measure. This is rather an expert defined scale which is then transferred and evaluated in a group of HSP patients. In order to gather items which are reported by patients or caregivers a structures process like a delphi process would be suitable.

The external judges (line 182) and clinical experts (line 182) should be defined. Are all neurologists (residents, trainees), movement disorder specialists? How many years of clinical experience with HSP patients do they have? How many outpatients are seen on regular bases (per week/months/year)? Are the experts trained with the SPRS and the other used scales/scores?

The introduction contains a big variety of methodological information. Please separate this and stick to the common structure using Introduction, method, and results instead of merging all those things already into the introduction (e.g. line 130-140 and 140-146). The sentence in line 123 is sufficient for the introduction, the rest is from my point of view methodology.

I disagree that further non-motor symptoms (line 158) do not affect the walking ability of the patients. Especially bladder impairment and urge affect directly the spasticity of patients. A “full bladder” causes increasing gait troubles (personal experience of hundreds of consulted HSP patients) with an increase in spasticity and a reduction in gait pace and distance. The other aspect not addressed in this study is depression or depressive reactions with regards to the HSP. I think that this clearly affects activity levels and endurance or just motivational aspects of the patients. This should be addressed.

I disagree that retaking a test on the next day helps to improve the validity of the scale (ll 221-223). In large cohorts (e.g. MoCA test) the retest-validity of tests was explored. The patients were outpatients, is this correct? Did they sleep in their used environments or did they stay over night at the hospital for the second assessment? These environmental changes with changes in sleeping behavior might already affect the “state” at day 2. Why did healthy subjects filled it only once? This is not a good control group if parameters are changed between those groups – Please discuss this in depth in your discussion.

The methodological sections is lacking the following information:

- How do you classify pure vs. complicated?

- Please state the molecular diagnosis including the mutation you found (interesting for the pure vs. complicated part)

- How was a clinical diagnosis of HSP set? Which exclusion examinations were performed prior to the diagnosis of HSP (if no gene was confirmed)?

Your test was probably in Italian, am I correct? Can you provide the original Italian Questions? I see major translation troubles if you “introduce” an English scale without validation in a native English cohort.

Minor:

- To my knowledge the HSP term was introduced and described by Anita Harding or Strümpell-Lorrain (the German Adolph Strümpel and the French Maurice Lorrain). I would change the first reference to either of those instead of Fink and colleagues. If you would like to cite a good review paper – please use a more recent paper than this 20 year old outdated review paper.

- I disagree with the term “low resistance” – the already in the manuscript suggested term “reduced or limited endurance” is better feasible. But I am not totally convinced that endurance is reduced after a resting phase an exercise is conceived with increased spasticity in the beginning phase but then with reduced spasticity and ongoing exercise I am unsure if this really leads to limited endurance or just because gait speed is reduced to a lower distance covered over time. From questionaires (compare the non-motor study data from Rattay et. al.) the reported maximum gait distance at self-chosen speed is not dramatically reduced compared to healthy controls.

- Averagely is a not commonly used English term. Maybe use “on average” instead (ll 218)

- The control group is not well chosen if the mean age differs by 10 years.

-

6. PLOS authors have the option to publish the peer review history of their article (what does this mean?). If published, this will include your full peer review and any attached files.

Reviewer #1: No

Reviewer #2: No

---

## [Author Response · Author response to Decision Letter 0]

24 Jan 2024

Dear Editor,

We thank you and the reviewers for your valuable feedback. 

We carefully read all the reviewers’ comments and addressed them below. 

- Journal Requirements:

Please provide an amended statement that declares all the funding or sources of support (whether external or internal to your organization) received during this study. 

“This work was supported by Fondazione Cariplo and Regione Lombardia (EMPATIA@Lecco project) and by the Italian Ministry of Health (RC2022 to E. Biffi). There was no additional external funding received for this study.”

We note that you have stated that you will provide repository information for your data at acceptance. Should your manuscript be accepted for publication, we will hold it until you provide the relevant accession numbers or DOIs necessary to access your data

Currently, our data are uploaded but restricted on ZENODO. We will modify the accessibility as soon as our paper will be accepted for publication. If you want to take a look on our database before the publication, we are willing to send you all data. 

- Reviewer #1

1. First, „development” of the questionnaire seems overstated since it had been developed before and was evaluated (and not modified) in this manuscript. My suggestion is to omit this term from the title, abstract and main text.

We agree with the reviewer and we have removed “development” from the title.

2. In addition, the term HSP-SNAP does not reflect that it probably does not reflect motor-independent symptoms like sensory symptoms, bladder dysfunction etc. which should be stated in the abstract.

We specified in the Abstract that our questionnaire collects information only on motor symptoms.

3. Abstract: The „Medical Outcome Survey Short Form“ should be specified by „(SF-36)“. State absolute range of the proposed HSP-SNAP in the abstract. Note the language of the questionnaire.

We modified the Abstract as requested.

4. Data availability statement: data are not freely available, but have to be requested via the given repository. This should be mentioned in the statement.

Currently, our data are uploaded but restricted on ZENODO. We will modify the accessibility as soon as our paper will be accepted for publication. 

5. Study registration in ClinicalTrials is credited, but a closer look shows that the study design was posted 1.5 years after FPI; this limitation must be mentioned in the methods as well as any changes to the protocol during the study.

We are aware that the trial was registered after the first patient recruitment but the delay was due to a change in our Institute’s policy. We mentioned this in the Methods section. However, no changes were done to the protocol during the study. 

6. An average of 10 minutes to complete a 12-item-questionnaire seems odd. This might have been caused by „exhaustive explanations by the evaluator“ which however may introduce substantial bias and must be discussed.

We thank the reviewer for this comment. We tried to clarify the time for the administration of the questionnaire and the potential bias as follows: 

Page 9, lines 205-208:” The HSP-SNAP was self-administered via paper and pencil. No time constraint was given to the patients. Participants received a clear explanation by the evaluator and could ask for clarifications. The evaluator used standardized instructions to avoid bias introduction. Furthermore, we recorded potential difficulties in understanding and completing each single item”. 

7. The study included pediatric and adult patients, with an inclusion starting at age 8. Given the clinical differences of pediatric onset HSP and the differences in PROM responses provided by pediatric cohorts, the authors should precisely state the ratio of minors and provide results of their statistical analyses when including adults only as a supplement.

As observed by the reviewer, we included in our study both pediatric (from age 9, according to our inclusion criteria) and adult patients. Our institute indeed regularly follows pediatric and adult HSP patients. Our aim was to validate a tool usable from 9 years old, which is the age at which a patient can provide his/her subjective perception of symptoms. Furthermore, our reference outcome for HSP, i.e. SPRS, was validated for children older than 9 years old, and we wanted to use the same approach. 

We verified that the results did not change if we include only the adult population. We report here these results for the reviewer.

The ICC of HSP-SNAP including only adults (N=35) was equal to 0.97, even higher than the one computed for the whole group, as expected.

The mean value of HSP-SNAP for the adult HSP group (N=35) and the adult healthy group (N=35) were equal to 20.3 ±7.2 and 42.5± 6.7, respectively, a couple of point smaller than the results obtained before. The means significantly differed as in the entire group (U=28.5, p<.001). This result is expected since we previously demonstrated that the age negatively correlates with HSP-SNAP score. 

However, we do not believe that adding these results in the supplemental material would improve the quality of the manuscript. Indeed, the validation of HSP-SNAP was done in a population of patients with HSP older than 9 and can be used in the future in this group.

8. Ref. 18 seems out of context. 

Ref 18 supports the use of PROMs as outcomes in clinical trials for many rare diseases and, in this case, for patients affected by Osteogenesis Imperfecta. We think that this reference should be cited in the background even if it includes a different population with respect to the one included in the current manuscript.

9. Ll. 131-146 do not belong to the „background & purpose“ section – they should be replaced by a much more concise summary of overall hypothesis and findings.

We thank the reviewer and agree with him/her. Therefore, we shortened the Introduction. 

10. „Unbalance“ should be rephrased throughout the manuscript.

We checked all the manuscript and rephrased with the correct term “imbalance”.

11. L. 278: please provide the reference.

The Italian version, administered to judges and patients, is provided as Supplement 1.

12. It is surprising that expert clinicians were different for assessing comprehensibility and content validity, and a reason should be provided. For the „expert clinicans“ group, year of experience with HSP patients should be reported.

We thank the reviewer for this comment that allowed us to clarify the description of our judges population. We integrated these data in the text. For assessing comprehensibility we selected 10 physical therapists that have been working in our Institute on average for 15 years. For assessing content validity we selected medical doctors (specialized neurologists, specialized physiatrists) and physical therapists with a specific and wide experience on our HSP patients and trained on disease specific scales. They have been working on neurological degenerative disorders for 14 years on average.

13. Table 2 lacks SD data (as indicated in the first row), and SD data should also be reported for „age at onset“.

We added SD data to Table 2.

14. The „COSMIN Reporting guideline for studies on measurement properties of patient reported outcome measures“ provided as a supplement is the unmodified raw document by PMID33818733. For each of the items on this list, the authors must provide a statement how this item was met in the present study.

We filled the Cosmin checklist form and uploaded it as Supplement 2. We added this sentence in the Study Protocol section at page 9, lines 185-187: “We designed our protocol following COSMIN guidelines for studies on measurement properties of patient reported outcome measures (Supplement 2)”.

15. The Italian questionnaire version that was actually applied should be provided as a supplement.

The Italian version, administered to judges and patients, is provided as Supplement 1.

16. The figure legends should be added to the submission file, including abbreviations.

We added figure legends in the manuscript.

17. The exact version of the SF-36 questionnaire should be specified. Moreover, the method of calculating SF-36 values for correlational analyses is unclear (physical or mental component scores? type of calculation). 

Thanks for this comment. We applied the Italian version of Apolone et al. (1997), translation of the original English version by Ware and Sherbourne (1992). We specified it in the text. We calculated SF-36 scores according to the User Manual published by Ware in 1993. We corrected the text specifying that we obtained a moderate correlation with both physical and mental component of the SF-36. We added also methodological details about the SF-36 in the methodological section.

18. l. 329: Define whether SD or SEM is reported here.

We used the standard deviation. This is now detailed in the text.

19. Fig. 2: The calculation of the x-axis value should be made clear to the reader in the figure legend. The color coding presentation of significance is unusual and not distinguishable on printing b/w. The smileys suggest that they were present on the actual questionnaires.

We thank the reviewer for this comment that allowed us to clarify the manuscript. 

In the Methods section we have clarified how the dimension scores are computed, i.e. “The dimension score is the mean value between the two items related to that dimension.” Furthermore, Figure 2 has been revised thoroughly, using both colour code and pattern code (for b/w printings) and removing smileys. The name of two items, i.e. imbalance and reduced endurance, were modified according to changes in the text of the manuscript. We hope the figure is more readable.

20. Typos must be corrected here: l. 84, l. 110, l. 114, l. 118 (twice), l. 119 (twice), l. 139, l. 286, l. 365, l. 408

We corrected the typos in the manuscript.

- Reviewer #2:

I see severe methodological issues with this “patient reported” outcome measure. This is rather an expert defined scale which is then transferred and evaluated in a group of HSP patients. In order to gather items which are reported by patients or caregivers a structures process like a delphi process would be suitable.

After suggestion of the Reviewer #1, we decided to focus our study on the validation of the Italian Version of the HSP-SNAP. Concerning the development of the questionnaire, we agree with the Reviewer #2 and we indeed stated this issue in the limitation section (“We thus decided in 2018 to develop the “Hereditary Spastic Paraplegia-Self Notion and Perception Questionnaire” (HSP-SNAP) designing the questionnaire with clinicians with a long-standing knowledge of patients’ perspective. The end users were involved in its comprehensibility and content validity assessment”) of our manuscript. We are aware that a Delphi process would have made stronger our questionnaire and work. However, after the expert definition of the questionnaire, patients were kept involved in the process. Notably, no patient reported the incompleteness of the questionnaire or the assessment of worthless items.

The external judges (line 182) and clinical experts (line 182) should be defined. Are all neurologists (residents, trainees), movement disorder specialists? How many years of clinical experience with HSP patients do they have? How many outpatients are seen on regular bases (per week/months/year)? Are the experts trained with the SPRS and the other used scales/scores?

We thank the reviewer for this comment that allowed us to clarify the description of our judges’ population. We integrated these data in the text.

For assessing comprehensibility, we selected 10 physical therapists that have been working in our Institute on average for 15 years. For assessing content validity, we decided to select medical doctors (specialized neurologists, specialized physiatrists) and physical therapists with a specific and wide experience on HSP patients and trained on disease specific scales. They have been working on neurological degenerative disorders on average for 14 years.

As a hospital with a Unit dedicated to rare disease of the central and peripheral nervous system, we see as inpatient and /or outpatient about 60 HSP per year. 

The introduction contains a big variety of methodological information. Please separate this and stick to the common structure using Introduction, method, and results instead of merging all those things already into the introduction (e.g. line 130-140 and 140-146). The sentence in line 123 is sufficient for the introduction, the rest is from my point of view methodology.

We thank the reviewer for the comment. We considerably reduced the Introduction saving the Background, the Aims and the Attended Results following more strictly the COSMIN guidelines. 

I disagree that further non-motor symptoms (line 158) do not affect the walking ability of the patients. Especially bladder impairment and urge affect directly the spasticity of patients. A “full bladder” causes increasing gait troubles (personal experience of hundreds of consulted HSP patients) with an increase in spasticity and a reduction in gait pace and distance. The other aspect not addressed in this study is depression or depressive reactions with regards to the HSP. I think that this clearly affects activity levels and endurance or just motivational aspects of the patients. This should be addressed.

I understand the point risen by the reviewer. I agree that bladder incontinence and urinary urgency are common symptom of HSP patients and that they impact negatively on patient’s quality of life. However, when we developed HSP-SNAP we were interested in exploring only motor symptoms related to walking. In the current manuscript we aim only at assessing reliability and validity of the HSP-SNAP questionnaire, therefore we eliminated the sentence:” Other non-motor symptoms reported in the survey by Rattay and colleagues [32] were not included since they do not impact on gait ability and performance despite their relevance”. We will take in to consideration the reviewer suggestion in future work. Concerning mood and/or anxiety disorders, we reported that 3 out of 40 patients were taking antidepressant drugs but we are aware that a more detailed description of mood disorders could have strengthened our analysis. We report the limitation written in the Discussion at page 19, lines 418-420:” Additionally, we did not collect specific information on mood or anxiety disorders potentially influencing the assessment and we assumed that the presence of depressive symptoms or symptoms of anxiety could be correlated with systematically lower scores of the instrument”.

I disagree that retaking a test on the next day helps to improve the validity of the scale (ll 221-223). In large cohorts (e.g. MoCA test) the retest-validity of tests was explored. The patients were outpatients, is this correct? Did they sleep in their used environments or did they stay over night at the hospital for the second assessment? These environmental changes with changes in sleeping behavior might already affect the “state” at day 2. Why did healthy subjects filled it only once? This is not a good control group if parameters are changed between those groups – Please discuss this in depth in your discussion.

We kindly thank you for your comment but we would like to underline the conceptual differences between reliability and validity in research projects.

Content validity assesses whether a questionnaire represents all relevant aspects to be measured whereas reliability, as defined by Cosmin, is “the extent to which scores for patients who did not change are the same for repeated measurement under several conditions”. Test–retest reliability of an instrument is computed by measuring subjects at two distinct occasions on the instrument and then computing the correlation. If the correlation is large, this is evidence for good test–retest reliability. 

Our patient were both in and outpatients and they slept in the same bed before the administration of the questionnaires. We ensure no environmental changes occurred in the assessment.

We tested reliability and validity of the HSP-SNAP questionnaire in the HSP population and then we collected “healthy” subject scores to define normative data. This is the reason why test-retest reliability was not assessed in healthy subjects.

The methodological sections is lacking the following information:

- How do you classify pure vs. complicated?

We classified patients according to symptoms presented at the evaluation following the indications stated by Appleton RE et al. We clarified this point in the text.

 - Please state the molecular diagnosis including the mutation you found (interesting for the pure vs. complicated part)

The molecular diagnosis, including mutations, was already reported in Participant description. We better specified which were pure or complicated within the text as follows: “Particularly, 10 patients were carrying mutation in SPG4 (9 pure and 1 complicated); 4 in SPG7 (complicated); 2 in SPG3A (pure); 2 in SPG11 (complicated); 1 in SPG5 (pure); 1 in SPG8 (complicated); 1 in SPG30 (pure); 1 in SPG31(pure); 1 in SPG35 (complicated); 1 in SPG72 (pure)”. 

- How was a clinical diagnosis of HSP set? Which exclusion examinations were performed prior to the diagnosis of HSP (if no gene was confirmed)?

We thank for the comment and we clarified the description in the Methods section as follows: “… 2) not genetically determined paraplegia with the exclusion of all other causes of spasticity through brain and spinal cord imaging studies, metabolic analysis, clinical history data (i.e. pregnancy and or/ delivery complication, Apgar evaluation at birth”.

Your test was probably in Italian, am I correct? Can you provide the original Italian Questions? I see major translation troubles if you “introduce” an English scale without validation in a native English cohort.

Thanks for this comment. Yes, the text that we validated was in Italian. We modified the title underlining this point. Furthermore, we modified the text as follows:

Page 6, lines 143-146:” Tab.1 shows, as a reference, the HSP-SNAP translated in English by a native speaker to be understandable by international users. A proper translation process including translation and then back-translation by an independent group was performed for the English version. The Italian version, administered to judges and patients, is provided as Supplement (S1_Table)”.

Minor:

- To my knowledge the HSP term was introduced and described by Anita Harding or Strümpell-Lorrain (the German Adolph Strümpel and the French Maurice Lorrain). I would change the first reference to either of those instead of Fink and colleagues. If you would like to cite a good review paper – please use a more recent paper than this 20 years old outdated review paper.

We cited a newer review of Meyyazhagan et colleagues (2022).

- I disagree with the term “low resistance” – the already in the manuscript suggested term “reduced or limited endurance” is better feasible. 

We modified the term “low resistance” in “reduced endurance”.

But I am not totally convinced that endurance is reduced after a resting phase an exercise is conceived with increased spasticity in the beginning phase but then with reduced spasticity and ongoing exercise I am unsure if this really leads to limited endurance or just because gait speed is reduced to a lower distance covered over time. From questionnaires (compare the non-motor study data from Rattay et. al.) the reported maximum gait distance at self-chosen speed is not dramatically reduced compared to healthy controls.

We would like to specify that what we ask to the patient is a personal point of view on his/her usual gait endurance, an estimation of performance not of capacity, according to ICF definition. In our experience HSP patients report to have reduced endurance.

- Averagely is a not commonly used English term. Maybe use “on average” instead (ll 218)

We thank for the correction.

- The control group is not well chosen if the mean age differs by 10 years. 

We report that the mean age differs by only 3 years with a P value of .337. HSP subjects (45.4 ± 17.8 years) and Healthy subjects (42.3 ± 19.6 years).

Thank you in advance for your attention.

Eleonora Diella and Emilia Biffi on behalf of all the authors

---

## [Decision Letter · Decision Letter 1]

26 Feb 2024

PONE-D-23-28019R1Validation of the Italian Version of a Patient-Reported Outcome Measure for Hereditary Spastic ParaplegiaPLOS ONE

Dear Dr. diella,

Thank you for submitting your manuscript to PLOS ONE. After careful consideration, we feel that it has merit but does not fully meet PLOS ONE’s publication criteria as it currently stands. Therefore, we invite you to submit a revised version of the manuscript that addresses the points raised during the review process.

We look forward to receiving your revised manuscript.

Kind regards,

Abeer El Wakil, PhD

Academic Editor

PLOS ONE

Journal Requirements:

**Additional Editor Comments:**

The manuscript has been greatly improved; however, some concerns are still pending and need actions.

Reviewers' comments:

Reviewer's Responses to Questions

**Comments to the Author**

1. If the authors have adequately addressed your comments raised in a previous round of review and you feel that this manuscript is now acceptable for publication, you may indicate that here to bypass the “Comments to the Author” section, enter your conflict of interest statement in the “Confidential to Editor” section, and submit your "Accept" recommendation.

Reviewer #1: (No Response)

Reviewer #2: All comments have been addressed

2. Is the manuscript technically sound, and do the data support the conclusions?

Reviewer #1: Yes

Reviewer #2: Yes

3. Has the statistical analysis been performed appropriately and rigorously? 

Reviewer #1: Yes

Reviewer #2: Yes

4. Have the authors made all data underlying the findings in their manuscript fully available?

Reviewer #1: No

Reviewer #2: Yes

5. Is the manuscript presented in an intelligible fashion and written in standard English?

Reviewer #1: No

Reviewer #2: Yes

6. Review Comments to the Author

Reviewer #1: The authors have adressed most of my suggestions and concerns. It is imperative, though, to clearly state this study's limitations in the abstract, i.e. that the scale is only validated in Italian at this point and that it does not specifically inquire about non-motor symptoms.

"Italina" should be corrected.

Reviewer #2: Dear authors,

the paper significantly improved throughout this review process.

Thank you for the hard work!

There are only three points left:

1) I disagree with the quotation of reference 18 as stated by reviewer one (I was reviewer #2) so therefore we both disagree and this should still be changed.

2) The information about the testing adults/children should be added to the supplement. I also follow the suggestions of reviewer #1.

3) With the discussion of the authors in this paragraph I am delighted.

[...] But I am not totally convinced that endurance is reduced after a resting phase an exercise is conceived with increased spasticity in the beginning phase but then with reduced spasticity and ongoing exercise I am unsure if this really leads to limited endurance or just because gait speed is reduced to a lower distance covered over time. From questionnaires (compare the non-motor study data from Rattay et. al.) the reported maximum gait distance at self-chosen speed is not dramatically reduced compared to healthy controls. [...]

I think this is very intersting point that should be added to the manuscript because this is a ongoing discussion in the field of HSP.

7. PLOS authors have the option to publish the peer review history of their article (what does this mean?). If published, this will include your full peer review and any attached files.

Reviewer #1: No

Reviewer #2: No

---

## [Author Response · Author response to Decision Letter 1]

8 Mar 2024

Dear Editor,

We thank you and the reviewers for your valuable feedback. 

We carefully read all the reviewers’ comments and addressed them below. 

Editor Comments:

The manuscript has been greatly improved; however, some concerns are still pending and need actions.

We thank the Editor for this comment. We have considered the additional concerns in this revision.

Reviewers' comments:

Reviewer #1: The authors have addressed most of my suggestions and concerns. It is imperative, though, to clearly state this study's limitations in the abstract, i.e. that the scale is only validated in Italian at this point and that it does not specifically inquire about non-motor symptoms.

We thank the reviewer for this comment. We have corrected the typo and we have added the limitations of the study in the abstract and as follows:

“Conclusion: Although HSP-SNAP does not investigate non-motor symptoms and we validated only its Italian version, it showed good validity and reliability and it could be used in combination with other objective outcome measures for clinical purposes or as endpoints for future clinical rehabilitation studies.”

Reviewer #2: Dear authors, the paper significantly improved throughout this review process.

Thank you for the hard work!

Thanks a lot for your comment and work.

There are only three points left:

1) I disagree with the quotation of reference 18 as stated by reviewer one (I was reviewer #2) so therefore we both disagree and this should still be changed.

According to your suggestion, we removed reference 18.

2) The information about the testing adults/children should be added to the supplement. I also follow the suggestions of reviewer #1.

We thank the reviewer for this comment. We added in the Methods and Results section (Supplementary material S3) the analysis on the group of adults.

3) With the discussion of the authors in this paragraph, I am delighted.

[...] But I am not totally convinced that endurance is reduced after a resting phase an exercise is conceived with increased spasticity in the beginning phase but then with reduced spasticity and ongoing exercise I am unsure if this really leads to limited endurance or just because gait speed is reduced to a lower distance covered over time. From questionnaires (compare the non-motor study data from Rattay et. al.) the reported maximum gait distance at self-chosen speed is not dramatically reduced compared to healthy controls. [...]

I think this is very interesting point that should be added to the manuscript because this is an ongoing discussion in the field of HSP.

We thank the reviewer for this fruitful discussion. We have added the following considerations to the Discussion section underling the openness of the point:

“However, we are aware of the ongoing debate in the field whether HSP’s gait is actually characterized by reduced endurance or just by reduced walking speed. This is a crucial open point that need further in-depth analysis. Besides, we can’t ignore that fatigue and pain, as reported by Rattay et al [38], are frequent symptoms that need to be screened and treated because they affect health-related quality of life”.

Thank you in advance for your attention.

Eleonora Diella and Emilia Biffi on behalf of all the authors

---

## [Editor Report · Decision Letter 2]

17 Mar 2024

Validation of the Italian Version of a Patient-Reported Outcome Measure for Hereditary Spastic Paraplegia

PONE-D-23-28019R2

Dear Dr. Diella,

We’re pleased to inform you that your manuscript has been judged scientifically suitable for publication and will be formally accepted for publication once it meets all outstanding technical requirements.

Kind regards,

Abeer El Wakil, PhD

Academic Editor

PLOS ONE
---

## [Editor Report · Acceptance letter]

24 Mar 2024

PONE-D-23-28019R2 

PLOS ONE

Dear Dr. Diella, 

I'm pleased to inform you that your manuscript has been deemed suitable for publication in PLOS ONE. Congratulations! Your manuscript is now being handed over to our production team.

Kind regards, 

on behalf of

Professor Abeer El Wakil 

Academic Editor

PLOS ONE